# Access to Geriatric Disability Care in India: A Roadmap for Research

**DOI:** 10.3390/ijerph191610018

**Published:** 2022-08-14

**Authors:** Priyadarshini Chidambaram, S. D. Sreeganga, Anupama Sanjeev, Sarah Shabbir Suwasrawala, Suman Gadicherla, Lalitha Krishnappa, Arkalgud Ramaprasad

**Affiliations:** 1Department of Community Medicine, Ramaiah Medical College, Bengaluru 560 054, Karnataka, India; 2Ramaiah Public Policy Center, School of Social Sciences, Ramaiah University of Applied Sciences, Bengaluru 560 054, Karnataka, India; 3Management-Academics, Higher Education, Atria University, Bengaluru 560 024, Karnataka, India; 4Department of Information and Decision Sciences, University of Illinois at Chicago, Chicago, IL 60607, USA

**Keywords:** geriatric, disability, access, healthcare, India, ontology

## Abstract

This paper presents an ontological review of the global research on access to geriatric disability care and a roadmap for future research to address the problem in India. First, the dominant research focus is on resources (human, financial, and spatial) that affect access to disability care; there is little focus on informational and technological resources. Second, functional disabilities are the dominant focus of the research, followed by cognitive, mental, and locomotor disabilities; there is little focus on speech, hearing, and visual disabilities. Third, barriers, inhibitors, and catalysts of physical access are the dominant focus, with relatively less focus on virtual access; there is very little emphasis on the drivers to access. Fourth, the primary, although not dominant, focus is on access for urban and rural populations; there is very little focus on access for underserved and indigenous populations. Future research must address these gaps systematically to improve access. This paper adds: (a) a systemic framework for the study of an important, complex, emerging problem; (b) a systematic review of the global research on the problem; and (c) a research roadmap to address the emerging problem in India.

## 1. Introduction

The world population is aging rapidly. The older population has overtaken the child population below 5 years in 2020 and is set to overtake the child population below 10 years by 2030 [1,2]. An increase in the proportion of older persons from 11% to 22% is expected between 2000 and 2050 [1]. (a) Western countries such as the UK and the USA are being strained in providing health and social care services for their older population; an increase in the older population in low- and middle-income countries can be more challenging [3]. India can learn from the global research on access to geriatric disability care in developing its own roadmap for research.

India has been undergoing a significant demographic transition in the past 50 years. It has resulted in a tripling of the population over the age of 60 years (i.e., the elderly) [4]. As per Census 2011, 8.5 percent of India’s population is geriatric amounting to 103 million people [5]. The United Nations Development Programme (UNDP, New York, NY, USA) estimates that the proportion is expected to rise to 19 percent in 2050. India’s life expectancy at birth, currently 67.9 years, is expected to reach 74 years by 2050 and will add to its geriatric population [6].

As India’s population ages, the country will face a shrinking pool of productive age groups and resources to support the elderly population’s healthcare. In urban areas, the changing societal Indian structure, such as urbanization, youth migration, and working women, is leading to the disintegration of the joint family system leading to a lack of care and support for the elderly [7]. A larger proportion of the elderly in India live in rural areas (67.1%) and face greater challenges in access to healthcare [8]. The elderly in rural India have limited access to healthcare in general and almost nonexistent long-term care except for untrained and informal care provided by family members. Thus, dramatic demographic, social, and economic shifts in both urban and rural India have created an urgent need for primary geriatric care services with an emphasis on preventive and disability care.

Most of the disabilities of the elderly are preventable and correctable. Timely and appropriate care can make the elderly more functional and thus improve their quality of life. While the accessibility of services for the elderly is an issue, disability is an important barrier to their accessing care. Other than the Ministry of Health’s recently introduced National Programme on Health Care for the Elderly (NPHCE), population aging is not a priority area, and the Government of India is yet to recognize its potential to become a major economic and health challenge [9].

With rising longevity, multiple morbidities, co-morbidities, and disabilities have become progressively common in older persons. While the disability rate across all ages in India was 2.2 percent, the elderly (60+ years) disabled constituted 21 percent of the total [10]. NSSO estimates that about 64/1000 and 55/1000 elderly suffer from one or more disabilities in rural and urban areas, respectively [11].

Hence, ensuring access to geriatric disability care will continue to increase in importance in India. The scale of this need will be very large given the size of India’s population, the scope of disabilities to be covered will be very wide given the diversity of its population, and the spread of the need will be very large given the geography of the country. There is not enough systemic understanding of the growing need for geriatric disability care, globally and in India. Consequently, the policies and practices for such care do not systematically address the emerging challenges. This study maps the existing global literature on access to geriatric disability care using an ontological framework adapted from our earlier publications on access to healthcare. The paper also highlights the emphases and gaps in the research which will help to develop a roadmap for research to improve access to geriatric disability care in India.

## 2. Materials and Methods

The ontology of access to geriatric disability care defines the dimensions of the challenge, the elements of each dimension, and together the boundaries of the challenge. It is a hierarchical representation and visually represents the complex combinations of pathways (lateral and associated) to address the challenge. The visualization is in structured natural English, making it easy to read and understand by experts and novices. It is like a ‘Google Map’, using which one can see all the ‘roads’ to address the challenge. The present ontology has evolved from earlier ontologies of access to healthcare [12] and access to healthcare during COVID-19 [13]. The development and application of these ontologies follow the description of the logic and process by Ramaprasad and Syn [14,15].

### 2.1. Ontology of Barriers to and Facilitators of Access to Geriatric Disability Care

Geriatric disability may be related to a person’s functional performance, locomotion, hearing, speech, seeing, mental faculties, and cognitive capabilities. These are listed in the ‘Disability’ column of the ontology in Figure 1. The object of disability care includes preventing the disabilities, promoting their care, curing both acute and chronic ones, rehabilitation, and palliation. These are listed in the ‘Type’ column of the ontology. If we combine the two columns, there are 6 × 7 = 42 potential types of disability care. They include: (a) functional disability care, (b) curative–acute mental disability care, and (c) palliative locomotor disability care.

The disability care may be administered by a wide range of personnel listed in the ‘Personnel’ column. The list includes physicians (general, specialist, AYUSH), traditional healers, nurses, health workers/activists, Anganwadi workers, pharmacists, social workers, care providers, peers, family, community groups (self-help, youth, and other), rehabilitation workers, and oneself. The care may be administered individually or jointly by these personnel at a private, public, or an NGO-based healthcare facility. These are listed under the ‘Ownership’ column. Combining the four columns, there are 3 × 6 × 7 × 17 = 2142 pathways to rendering geriatric disability care. They include: (a) access to private functional disability care by nurse, (b) access to public curative–acute mental disability care by physician–specialist, and (c) access to NGO palliative locomotor disability care by care provider.

The ‘Access’ to these pathways to geriatric care may be physical or virtual; they are listed in the corresponding column. The ability to access the variety of care will depend on the interplay of four types of ‘Forces’, namely: barriers to access, inhibitors of access, catalysts of access, and drivers of access. These are listed in the corresponding column of the ontology. These forces are derived from the seven types of ‘Resources’—spatial, temporal, financial, informational, human, technological, and infrastructural—and their subtypes are listed under the first column. Thus, there are 19 × 4 × 2 = 152 ways resources can affect access to geriatric disability care. They include: (a) spatial–distance barrier to physical access, (b) technological–IT catalyst of virtual access, and (c) financial–income driver of physical access.

The forces that affect access will also depend on the type of ‘Population’ listed in the last column. The urban, rural, peri-urban, underserved, and indigenous populations will likely have different pathways by which resources affect access to geriatric disability care. Thus, the spatial–distance barrier to physical access for urban populations, rural populations, and indigenous populations can be very different. The same would be true for technological–IT catalysts of virtual access and financial–income drivers of physical access.

In sum, the ontology encapsulates 2142 × 152 × 5 = 1,627,920 barriers to and facilitators of access to geriatric disability care. It (a) provides a unified framework to address the problem, (b) defines the elements, dimensions, and boundaries of the problem, and (c) encapsulates the combinatorial complexity of the problem. Three illustrative pathways are: (a) spatial–distance barrier to physical access to private functional disability care by nurse for urban older population, (b) technological–IT catalyst of virtual access to public curative–acute mental disability care by physician–specialist for rural older population, and (c) financial–income driver of physical access to NGO palliative locomotor disability care by care provider for underserved older population.

### 2.2. Method

The existing global literature was mapped onto the ontology to visually synthesize the state of the research on access to geriatric disability care. The monad map and the theme map were generated from the mapping. The maps visualize the landscape of the domain and highlight the barriers to and drivers of access to geriatric disability care.

The corpus of research was created by searching Scopus on TITLE-ABSTRACT-KEYWORDS of the articles indexed in the database. The search process and results, following the PRISMA [16] reporting guidelines are given in Figure 2. We experimented with several search terms. The final search was conducted on 12 April 2021, at 10:35 a.m. IST using the string TITLE-ABS-KEY (care AND access AND (geriatric OR elderly OR older) AND disability). It yielded 1367 documents. Scopus is a curated, comprehensive database that includes many medical journals including those in PubMed (Medline). However, it may exclude a few articles in some medical journals.

The resulting items included different document types such as reviews, notes, letters, conference papers, editorials, and other types of documents. In the first iteration, we excluded 307 non-journal articles resulting in 1060 articles retained for further screening. In the first stage of screening, 3 incorrect entries were excluded. In the second stage of screening based on relevance, 476 articles were excluded leaving 581 articles to check for eligibility. In the last stage, 6 articles without abstracts were excluded from the corpus. The remaining 575 articles were included in the final mapping.

The title, abstract, and keywords of the final 575 included articles were downloaded and imported into an Excel spreadsheet for mapping. The reference management software Zotero (Corporation for Digital Scholarship, Vienna, VA, USA) was used to store the corpus of included articles.

The final corpus of 575 articles was coded onto the ontology by four of the authors. The coding was performed in pairs. The corpus was divided into six approximately equal sets (corresponding to six pairs from the four coders) and assigned to each pair of coders. Each member of the pair coded the articles independently.

The pair of coders iterated between themselves until they arrived at a consensus. Post the individual rounds of coding by each coder, discrepancies, if any, were discussed and arrived at a consensus for the final coding. After a preliminary round of coding, two elements highlighted in the corpus were added to the ontology. They are ‘infrastructure’ resource and ‘cognitive’ disability.

To ensure reliability and validity, each of the four authors iterated the coding two times. A glossary of dimensions and elements was used to ensure the validity of coding. Some definitions in the glossary were clarified after the preliminary round of coding. Only the dimensions and elements explicitly articulated in the title, abstract, and keywords were coded. Implicit elements were not coded. Binary (1 for present, 0 for absent) coding was used. The coding was not scaled or weighted. The presence and absence of information convey equally important information in the analysis.

All the coding was synthesized into one Excel spreadsheet for further analysis. It represents the synthesis of the corpus by the four authors coding in pairs, with multiple iterations and validation.

## 3. Results

The results of mapping the research corpus onto the ontology are presented through a monad map (Figure 3) and a theme map (Figure 4). They are described next.

### 3.1. Monad Map

The monad map in Figure 3 visually and numerically summarizes the frequency of occurrence of each dimension and element of the ontology. It is essentially a histogram (frequency chart) of each dimension and elements of the dimension. The histograms are presented in the order in the ontology.

The number adjacent to the dimension name and the element is the frequency of its occurrence in the 575 journal articles that were reviewed and mapped. The bar below each element is proportional to the frequency relative to the maximum frequency among all the elements. Since each item can be coded to multiple elements of a dimension, the sum of the frequency of occurrence of elements may exceed the frequency of occurrence of the dimension to which the elements belong. The monad map is described below.

The dominant focus of the articles is on the resources (537) followed by type of disability (334), access (275), force (263), type of care (251), personnel (211), population (103), and ownership (47). The articles cover a spectrum of resources that affect access to disability care. It is heavily focused on human resources, i.e., human other (306) and human sociological (195). In addition, spatial dwelling and financial income are highly emphasized. There is a medium focus on financial dependence (89), financial expenditure (75), infrastructure (68), spatial location (61), informational education (53), technological medical (51), human psychological (45), technological IT (44), temporal availability (37), spatial transportation (36), and cultural (32). There is little focus on informational stimulant (14), spatial distance (13), and temporal scheduling (3). There is no emphasis on technological transportation (0). Amongst disabilities, the articles dominantly focused on functional (197), cognitive (93), mental (83), and locomotor disability (78). There is less emphasis on visual (40) and hearing disability (23). The least focus is on speech disability (15). The type of access to care focuses mainly on physical access (248); there is less emphasis on virtual access (36).

The forces for care dominantly focus on barriers (119) and catalyst (98) and less on inhibitor (89). There is the least emphasis on drivers (31) of access to care. The type of care is focused primarily on rehabilitative (94), followed by curative chronic (83) and preventive care (60). There is some focus on curative acute (35) and promotive care (23) and least on palliative care (14). Among the personnel, the majority focus is on physician specialist (54), care provider (54), and physician general (52). There is a moderate focus on community other (36), family (33), self (28), and nurse (21). There is less emphasis on social worker (18), rehabilitation worker (11), community youth groups (5), health worker/activist (3), community self-help groups (3), traditional healer (2), physician AYUSH (1), and pharmacist (1). There is no mention of Anganwadi (0) and peer (0).

The focus on population segments is lesser than on other dimensions except for ownership, with a dominant focus on rural (58) and urban population (52). It is lesser on underserved (18) and peri-urban (11) and least on indigenous population (6). The articles focus least on ownership with a dominant focus on public (36), less on private (16), and least on NGO (5).

### 3.2. Theme Map

While the monad map highlights the emphases on the elements of the ontology, the theme map visually summarizes the co-occurrence of the elements of the ontology in the population of articles. Some themes may be inferred intuitively from the monad map. For example, elements that occur frequently are likely to co-occur. However, the theme map formally computes and represents these co-occurrences, as described below.

Hierarchical cluster analysis was performed using SPSS (Statistical Package for Social Sciences; IBM, Chicago, IL, USA) with the simple matching coefficient (SMC) as the distance measure and the nearest-neighbor aggregation procedure. The dendrogram so derived is an exact visualization of the co-occurrence of the elements in the corpus. The detailed rationale for the choice of the clustering method and the presentation of the results are given in La Paz et al. [17] and Syn and Ramaprasad [18]. The five themes represent the five equidistant clusters in the dendrogram of the agglomeration [18]. As with the interpretation of the results of any cluster analysis, the choice of the number of clusters is subjective. We standardized the method by choosing five equidistant clusters representing five themes. The colors in Figure 3 highlight the elements of the five themes.

The primary research theme (in red) is the role of general human resources in physical access to disability care. It is a short segment of many potential pathways for access to disability care in the ontology—human resource other and physical access to disability care. The potential pathways may be barriers, inhibitors, catalysts, or drivers; include private, public, or NGO care providers; and include one or more types of disability, personnel, and population. Further, among the 19 resources, it includes only one, and among the two types of access, it includes one. The primary research theme is two-dimensional and one-leveled—it is simple.

The secondary research theme (in brown) is the role of spatial–dwelling, financial–income, and human–sociological resources in access to functional disability care. It is three short segments of many potential pathways for access to disability care in the ontology. They are spatial–dwelling and access to functional disability care, financial–income and access to functional disability care, and human–sociology and access to functional disability care. These potential pathways may include different forces, ownerships, types of care, disabilities, personnel, and populations. Among the 19 resources, the theme includes three, and among the seven disabilities, it includes only one. The secondary research theme is also two-dimensional but two-leveled—it too is simple.

The tertiary research theme (in yellow) is the role of financial dependence as a barrier to and catalyst of access to cognitive disability care. It is two short segments of many potential pathways in the ontology—financial–dependence barrier to cognitive disability care and financial–dependence catalyst of cognitive disability care. The potential pathways may include different types of access, ownerships, types of care, personnel, and populations. It includes one among 19 resources, two of four forces, and one among eight disabilities. The tertiary research theme is three-dimensional and two-leveled—it too is simple.

The quaternary research theme (in blue) is the role of many resources (spatial–location, financial–expenditure, informational–education, human–psychological, technological–medical, and infrastructure) as inhibitors. They inhibit access to: (a) preventive, curative–chronic, and rehabilitative locomotive, and mental disability care, (b) by physician–general, physician–specialist, care provider, and community–other personnel, and (c) in urban and rural populations. It includes 288 long segments of many potential pathways in the ontology. The pathways may include different types of access and ownership. The theme is six-dimensional and many-leveled—it is complex.

The quinary research theme (no color) summarizes the absence in the research corpus. Spatial–distance and spatial–transportation are not part of any theme, although spatial–location as an inhibitor is part of the quaternary theme, and spatial–dwelling affecting functional disability care is part of the secondary theme. Neither temporal–availability nor temporal–scheduling is associated with the forces that affect access or the other dimensions. Similarly, research on the force of information as a stimulant, human culture, information technology, and transportation technology in affecting access are absent in the corpus. Among the forces, thematic focus on drivers of access is absent, although barriers to, inhibitors of, and catalysts of access are included. Similarly, thematic inclusion of virtual access is absent, although physical access is part of the primary theme. Ownership of the care provider is not a thematic factor in the research corpus—there is no systematic distinction between the private, public, and NGO care providers. The theme highlights the exclusion of systematic promotive, curative–acute, and palliative care of the elderly disabled and at the same time of their hearing, speech, and visual disabilities. It also highlights the exclusion from the studies of a large variety of personnel that can provide elderly disability care—AYUSH physicians, traditional healers, nurses, health workers/activists, Anganwadi workers, pharmacists, social workers, peers, family, community (self-help and youth groups), rehabilitation workers, and the self. It also highlights the absence of systematic consideration of the peri-urban, underserved, and indigenous populations in the research. The quinary theme is eight-dimensional (same as the ontology) and many-leveled—there is a vast, complex, unresearched domain.

Overall, the themes are in decreasing order of dominance in the research—the primary theme is the most dominant, and the quinary theme is absent. The three most dominant themes are simple; they are two- to three-dimensional and one- to two-leveled. The fourth theme encompasses more dimensions and layers; it is complex. The quinary (absent) theme is a vast complex domain across many dimensions and levels that is yet to be researched.

## 4. Discussion

The ontology-based analysis of the 575 research journal publications on access to geriatric disability care shows the selectivity and segmentation in the research. The monad map shows that the dominant focus of the articles is on the resources followed by type of disability, access, force, type of care, personnel, population, and ownership in descending order. While the research covers all the dimensions, the coverage of the elements under each dimension is not systematic. It neglects crucial elements that could play a key role in providing access to geriatric disability care. The emphases and gaps in each of the dimensions are discussed below.

### 4.1. Resources

Among the resources, distance and transportation are key to ensuring that the elderly can access disability care. The elderly who are homebound in a country like India are acutely in need of transport services or mobile units to be able to access care [19]. These topics find little or no mention in the identified research. However, the kind of dwelling (home, senior care home) is emphasized in the research. The availability of specific health services and scheduling visits with these services have not been addressed in the research corpus. Especially in India, there is a dearth of functioning facilities for the disabled elderly, and this needs to be addressed in terms of mobile units, hospices, and daycare centers [20].

There is only exploration of information being delivered to the elderly as educational. However, it does not explore information as behavior modifiers/stimulants. Technology has been studied only with reference to advances in medical equipment, aids and instruments, and technological assistance for the elderly in terms of information technology. There appears to be an assumption that the elderly may not be able to adapt to the recent technological advances. The availability or access to technology among the elderly is also a challenge considering many Indian households do not possess mobile phones or laptops, especially in rural areas [21]. The specific availability of technology and its use for virtual access to healthcare must be studied. Technology in terms of smart homes must also be studied to see how older persons can obtain well-being and disability-friendly care in the confines of their own homes instead of limited and expensive care in health facility settings [22].

Last, cultural resources are also missing in the research. Cultural competency in care delivery is another barrier that healthcare providers must be mindful of to provide services that are respectful of and responsive to the health beliefs, practices, and needs of older adults of diverse backgrounds [23,24].

### 4.2. Forces

A large emphasis is laid on the forces that enable or disable access to healthcare, but overwhelmingly, it is tilted toward resources/traits that incapacitate the elderly, and there is very little focus on what factors can work to drive the elderly to access healthcare [25,26]. For example, these drivers could include family, friends, and peers. They can be involved in the elderly’s social lives through activities at nursing homes. They can also provide psychological support for their mental health and help deal with the emotional difficulties arising from the disability [27]. Connectivity as a driver is another crucial mode of access to care. Delivering services to the doorstep via mobile health clinics eliminates many logistical barriers such as a long waiting time, scheduling appointments, and complex administrative processes [28]. There is a lack of qualitative studies which would best explain how enablers and barriers act in the provision of elderly healthcare.

### 4.3. Access

The research corpus seems to be almost entirely focused only on physical access to care. This is consistent with research as it is missing the study of technological resources for the elderly. Especially during the COVID-19 pandemic when there has been disruption of physical access to care, more research is needed on improving the availability and uptake of virtual care by older persons. With the push for Digital India and telemedicine, research regarding elderly disability in India must also focus on this area to find solutions to make technology accessible to all [29]. The limitation to technological access is also discussed in terms of resource constraints. If channelized appropriately in terms of providing mobile telemedicine units, it can help healthcare reach populations that are underserved and have physical inaccessibility to health facilities or specialists [28].

### 4.4. Ownership and Type of Care

The overall literature on the kind of facility where the elderly seek healthcare is sparse. It is the dimension that is the least emphasized in the corpus. In a country like India, where private healthcare costs are high, providing quality disability care through the public healthcare system supported by the non-governmental sector is vital [30]. Moreover, the type of care to be provided by these facilities is crucial.

Most of the disability care discussed in the literature is rehabilitative and chronic curative care followed by preventive care. Focus on palliative, acute curative, and promotive care is less emphasized. The World Health Organization (WHO, Geneva, Switzerland) has identified palliative care as essential for improving the quality of life among those afflicted with life-threatening illnesses, and the care is obtained by 14 percent of those who need it [31]. Health promotion to prevent secondary conditions has been a neglected area, promotive care to mitigate the effects of disabilities on functional and mental aspects can be achieved with community-based initiatives [32]. Palliative and promotive care has been addressed least in terms of the healthcare services being studied. With the commitment toward Universal Health Care in India through initiatives such as Ayushman Bharat, it is important to widen the scope of services offered to older persons, especially in the public sector.

### 4.5. Type of Disability

Among the disabilities that have been emphasized in the research, most have referred to functional disability due to chronic morbidities and aging causing difficulty in carrying out everyday activities. The high prevalence of activities of daily living (ADL) disabilities in lower-middle-income country settings ranging from 16.2 percent in China to 55.7 percent in the Indian population might be the reason for the focus on this aspect [33]. Many studies which mention chronic diseases have also often not overtly mentioned any functional disability [20] and hence were not mapped onto the ontology.

Mental disabilities among older people are frequently not seen in the healthcare setting as they are construed as being part of old age by family members. Research to improve access to care is required in this regard [34]. There is sweeping reference to the elderly being disabled, infirm, or frail without discussing specific types of disability. Hearing, speech, and visual disabilities, which are some of the widespread disabilities among older people, have been studied very little. With aging, cognitive decline, vision, and hearing deficits emerge. Coincidentally, the prevalence rate of all three types of impairments is rising [35]. Despite the high rate of comorbidity, these disabilities continue to be unrecognized, under-diagnosed, and under-treated [35]. Often, the disability arising from many of the specific impairments is considered to cause functional disability and therefore is considered under it [36]. The wholesome concept of functional ability, which is determined by the intrinsic capacity of an individual (i.e., both physical and mental), the environment in which they live (including physical, social, and policy environments), and the interactions among them, is not explored in the research on the geriatric population.

Research on geriatric disability should focus on correctly identifying, labeling, and quantifying the disability as there are other social benefits provided by the government for the disabled. It will help the health system assess/budget the resources needed to be allocated for elderly care.

### 4.6. Personnel

The kind of care providers is mostly physicians either general or specialist or general care providers. There is very little engagement in the research corpus on how the older population can be independent and self-reliant or be supported by trained local human resources such as community or trained auxiliary health staff for healthcare. Personnel such as nurses and self-help groups can implement interventions such as healthy aging classes. Through these, one can build on the intrinsic capacity that will address physical and mental capacities [37]. Rehabilitative care in homecare settings by family members has also not found evidence of superiority in resource-constrained settings such as in India. Developing rehabilitative systems with multidisciplinary teams that involves interactions with caregivers is becoming a critical model for geriatric disability care [38]. Further research into this can inform policy on how the overburdened and understaffed public and private systems in India can be effectively unburdened while also providing the necessary care to these individuals.

### 4.7. Population

The mention of rural and urban populations is limited to more of a description rather than comparing or setting the context of how these affect healthcare access for disability care. The study by Kalaiselvi et al. highlights how such population descriptions are merely used to describe the study setting [36] with few studies about the differences in access to healthcare among populations [39]. Access to healthcare for the elderly disabled from other settings such as underserved and indigenous settings is the least emphasized. The elderly with disabilities and who are underserved have reduced access to primary care and need specific policy priorities [40].

### 4.8. Thematic Emphases and Gaps

Thematically, the emphases and gaps highlight the segmented, selective, and siloed logic of the research. The primary theme highlights the role of human–other resources in physical access to disability care. It neglects the force the resource exerts, ownership of the care provider, type of care provided, disability cared for, personnel providing care, and population cared for. The bracket of human–other characteristics included various factors such as gender, age groups, ethnicity, race, occupation, education, and marital status. The research focus on these characteristics that affect physical access to care in terms of education, gender, and groups is one of the important nuances the research has covered. The research indicates older women have more unmet healthcare needs than their male counterparts [41]. The use of eHealth also depends on the human–other characteristics mentioned above. For instance, the usage of eHealth significantly depends on higher educational attainment, employment, and household income [42].

While the above factors are emphasized in the primary theme, it misses other critical resources that can be drivers of physical access to different types of care for disabilities which can be delivered by personnel for the geriatric population. For example, cultural resources can be improved to develop competence in providing physical care in a manner that creates respect for patient beliefs, to treat specific conditions, and to have customized interventions to modify behavior by healthcare providers [23]. This will act as a driver for physical access to care and, in addition, involve stakeholders such as caregivers, nurses, and physicians in providing care to the previously underserved geriatric population due to cultural mismatch. The study by the American Geriatrics Society Ethnogeriatrics Committee shows that culturally competent care is important for promotive care as well [23].

Physical access using its abundant human–other resources, such as paid and unpaid caretakers, has been the norm in India for disability care. These resources are easily available at a low cost. As the economy develops, the availability and cost of these resources will change and likely make such access costlier. The digitalization of care heralded by COVID-19 and current healthcare policies may provide a segue to change the emphasis, especially for disabilities that are amenable to tele-care, such as mental and cognitive disabilities.

The secondary theme highlights the role of spatial–dwelling, financial–income, and human–sociological resources in access to functional disability care. The resources highlighted in accessing care for functional disability are emphasized in the research, particularly regarding the community–dwelling geriatric population. For this segment of the population, the resource of having Medicare provides them the benefit of accessing care following the Medicare claims for functional impairments [43]. While the community–dwelling geriatric population having Medicare is provided with the benefit of accessing care, there might be barriers that different groups and people with specific disabilities face, for example, the location and availability of a physician for a person of color. The location of these beneficiaries could be a barrier for them in accessing Medicare due to non-availability in their neighborhood or at the person’s usual treating physician [44]. In addition, the research neglects to lay emphasis on mental disability and falls short in meeting their needs. Availability of care appears to fall short because of supply barriers as well as the lack of responsiveness of the providers toward the geriatric population with mental disabilities [44].

Spatial–dwelling, financial–income, and human–sociological forces have a significant bearing on the access to functional disability care in India. However, the characteristics of these forces must be interpreted in India’s context. The density, design, and development of these dwellings are different from those in the West, and within India, they are different between rural and urban areas; the financial income is generally lower; and the human sociology is very different and heterogenous across the country.

The tertiary theme emphasizes financial dependence as a barrier to and catalyst of access to cognitive disability care. Research indicates that financial dependence to access care through formal sector pensions has been a critical catalyst in providing social security to the geriatric population with disability [21]. In addition, the provision of assistive aids through technological inputs for the elderly is paving the way for large-scale social and technological developments for their health and well-being [21].

Though the theme covers the resource of financial dependence as a barrier and catalyst of access to cognitive disability care it ignores the role other resources play. It ignores the forces that could influence the type of access with the help of a provider in different settings. For example, several people are now inclined toward self-management interventions through technological advances. However, the geriatric population with cognitive disabilities has difficulties with self-managing their healthcare [45]. These barriers call for approaches that will support the integration of solutions involving patient–caregiver dyads as they can be the intermediaries between technology and the person seeking care [45].

In many traditional multi-generational families in India, financial dependence on the older generation may be a catalyst and a facilitator of cognitive disability care. On the other hand, for families with limited financial resources, the dependence may be a barrier.

The quaternary theme is less segmented and selective compared to the other themes. Though it is spread across most dimensions, it remains agnostic to two important dimensions, the type of access and ownership of the care provider. While it covers significant resources as inhibitors, it ignores the resources that could be a driver for virtual access to private care for disability care to the older population. For instance, given the challenges of the aging population and pandemics such as COVID-19, eHealth services have become integral for an inclusive, effective, and robust healthcare system. A study by Ali et al. suggests that with public and private efforts, there will be an increase in the availability of ICT infrastructure supporting the elderly population with disability [42]. Further, interventions are also being made in the oral healthcare system with models that target specific community organizations where the high-risk underserved geriatric population lives. The model includes collaborative on-site and clinic-based teams by establishing ‘Virtual Dental Homes’ [46]. This model not only includes dwelling as a resource, but it also includes location and availability to access care. It will enable different ownership of care providers to deliver better healthcare which has the potential to drive down total health care costs for older adults and people with disabilities [46].

Management of locomotor and mental disability care is likely to be more important in the individualistic countries of the West and less in the collective countries of India and the East. The impact of spatial–location, financial–expenditure, informational–education, human–psychological, technological–medical, and infrastructure resources as inhibitors of these types of disability care could be very different between the two types of society. For example, in individualistic countries, spatial–location (where they live) and technological–medical (monitoring) resources may be more important. On the other hand, informational–education (what to do) and human–psychological (motivation to do) resources may be more important in collectivist countries.

The quinary theme highlights significant systematic oversights in all dimensions. For example, in resources, crucial elements such as spatial–distance and spatial–transportation are not part of any theme. Neither are temporal–availability nor temporal–scheduling. Research does not focus on the barriers the elderly persons with disability face due to the lack of transportation for those that live alone. Further, studies have proved that older adults with disabilities living alone have elevated odds of delayed care [47]. Focus on the above-mentioned resources as drivers is missing in research and the ownership of the care provider does not appear in any theme. For example, providing access to care for the elderly who live alone needs interventions such as public healthcare programs that involve home and community-based care and non-emergency medical transportation [47].

Further, the quinary theme highlights the exclusion of promotive, curative–acute, and palliative care of the elderly disabled and at the same time of their hearing, speech, and visual disabilities. The lack of attention to promotive and palliative care of the elderly disabled and the absence of a defined taxonomy of disabilities have made the research selective. Hearing loss not only affects spoken language, but it also impacts the cognition and mental health aspects. Interventions such as early screening and access to cochlear implants have the potential to ameliorate this burden [48]. Access to palliative care is the least emphasized in the research and shows the lacuna in access to geriatric disability care. Palliative care at home via care provided by nursing homes are upcoming interventions to assist the geriatric population with disability [49]. Additionally, the exclusion of a large variety of personnel that can provide elderly disability care shows that the research on the topic is not systematic. Rehabilitative, palliative, and promotive care requires the integration of personnel to provide care. The WHO’s integrated approach to care could be adopted for geriatric disability care such that there is cooperation with auxiliary staff, such as social workers, medical assistants, and volunteers [50,51].

In a populous, collectivist country like India, the variety of personnel not emphasized in the research corpus could play a very important role. They include physicians–AYUSH, traditional healers, nurses, health workers/activists, Anganwadi workers, pharmacists, social workers, peers, family, community (self-help groups, youth groups), rehabilitation workers, and oneself.

In summary, the primary themes only discuss resources and physical access to disability care. Ownership of healthcare, whether it is public, private, or NGO, appears only in the quinary theme. The type of care, personnel, and population is present only in either quaternary or quinary themes. The primary focus of the research articles has been on the availability or lack of resources in accessing care for disability by the geriatric population mainly on human socio-economic factors (age structures, gender, caste, employment, race, income, location). This highlights intrinsic factors/personal characteristics and emphasizes less extrinsic or structural resources such as the availability of health services, infrastructure, IT, or transportation. Physical access to care is emphasized disregarding the geriatric population using virtual access. Especially in the COVID-19 pandemic when there has been disruption of physical access to care, more research is needed on improving the availability and uptake of virtual care by older persons.

## 5. A Roadmap for Geriatric Disability Care Research in India

Geriatric disability care is an important, growing challenge for India. In the following, we propose a roadmap for research to address the challenge based on: (a) the ontology that was proposed in the paper, (b) the analysis of the emphases and gaps in the global (including India-based) research on the subject, (c) limitations of the present research corpus, and (d) knowledge of conditions and needs particular to India. The research following the roadmap should become the input for translation into policies on and practices for geriatric disability care in India.

The ontology is a comprehensive framework to visualize a roadmap for geriatric disability care research in India. It includes all the dimensions, elements of the dimensions, transverse pathways across dimensions, and associative pathways within dimensions that could be included in the roadmap. We discussed the strengths and shortcomings of the present global research corpus within this framework. We highlighted the emphases on and gaps in the coverage of the dimensions, elements, and pathways in the ontology. In addition, we noted two major limitations of the present corpus: first, the absence of a standard definition of the elderly—sometimes the geriatric population is grouped with all adults above 18 years—and second, the absence of a standard taxonomy of geriatric disability, including disability caused by chronic diseases.

The taxonomy of geriatric disability included in the ontology is appropriate and sufficient for the present. New elements could be added as the domain evolves. For example, primordial care could become the first stage of geriatric disability care. The present elements can be refined using subcategories to obtain a fine-grained roadmap. For example, locomotor disabilities may be with respect to fine motor skills (for operating a keypad, for example) or coarse motor skills (for heating a cup of coffee, for example). The present taxonomy could be the foundation of the proposed roadmap. The prevalence of the different types of disabilities among the different populations will determine the priority given to each type of disability. Thus, the first three segments of the roadmap are to:Develop a national standard definition of the elderly for India.Formalize the taxonomy of geriatric disability for India.Estimate the different disabilities among the different populations in each state and union territory of the country. Healthcare is a joint responsibility of the Central and State/Union Territory governments. The estimates must be a collaborative effort of the governments.

The fourth segment is to define the potential role of different personnel in the different types of care for each disability. This can be represented as a simple Type × Personnel table for each disability. Research on these combinations could aid policymakers and practitioners in the following ways:Planning for the training and skill development of personnel listed in the ontology for different types of care for the various disabilities.Direction and deployment of the above personnel for effective and efficient care.

The training, skill development, direction, and deployment must be based on evidence about the present emphases and gaps. They must: (a) reinforce Type × Disability × Personnel × Population combinations that have been effective, (b) redirect the combinations that have been ineffective, and (c) research innovative combinations that would be effective.

The fifth segment is to describe the ownership mixes that could deliver the different types of care for the five population types physically and virtually. Given the emerging scale and scope of geriatric disability care in India, the Access × Ownership × Type combinations must be well differentiated and integrated to address the needs of the different populations. For example: (a) NGOs may be effective in providing virtual access to preventive locomotor disability care by a nurse for underserved older populations, (b) private facilities may be effective in providing physical access to rehabilitative mental disability care by a specialist physician for an urban older population, and (c) public facilities may be deployed to provide virtual speech disability care by a care provider for a peri-urban older population.

The sixth segment is to identify and develop strategies to counter the barriers and inhibitors and deploy the catalysts and drivers to manage the different resource forces that could affect the above pathways to care. In formulating these strategies, one must learn from the global research, policies, and practices but customize them to local Indian conditions. Local feedback and learning systems must be built into the implementation of these strategies to aid their adaptation and evolution for greater effectiveness.

## 6. Strengths and Limitations

This study provides a comprehensive framework for deconstructing a combinatorially complex problem. It provides a systemic framework for describing the problem and its dimensions, elements, and pathways. It defines the problem and its boundaries in easily understandable structured natural English. The framework can be applied to scientifically study the problem and apply the insights to develop appropriate policies and practices. It is a means to address a looming, large, emerging problem globally and in India. At present, there is no similar framework.

The framework was used for a systematic, ontological review of the global (including India-based) research on the problem. The review highlights significant emphases and gaps in the research literature; they need to be addressed to develop effective solutions to the problem. It highlights some basic domain issues—such as the absence of a taxonomy of disabilities. It also highlights many other issues based on the research coverage. We believe this is the first such comprehensive review.

The framework and the review together help develop a roadmap for research to address the emerging problem. It systematically highlights the pathways to be reinforced, those to be redirected, and those to be researched. A periodic review of the research using the framework could provide feedback to learn and redirect the trajectory of research and, consequently, policies and practices that translate the research. As of 23 July 2022, 191 more articles have been published on the subject. Such additions can be cumulatively added to a subsequent analysis. This too would be a unique contribution.

A limitation of the study is the focus on articles in Scopus to the exclusion of articles in medical journals excluded from Scopus. Scopus includes articles from PubMed (Medline) but not other medical databases. In the future, the study could be extended to these articles too.

A second limitation is the uneven distribution of the knowledge across countries and geographical regions. On the one hand, this may help the global–local knowledge transfer, in both directions. However, in the short term, it may also bias the definition and analysis of the problem toward the Western perspective.

A last limitation of the study is that it is based on formalized knowledge encapsulated in peer-reviewed journal articles. A lot of non-formalized, cultural, local, indigenous, traditional knowledge may be relevant to address the problem. They may not have been formalized simply because nobody has published papers on them. The framework and the method provide an opportunity to formalize such knowledge and use them to address the problem.

## 7. Conclusions

Access to geriatric disability care is a significant problem in India and globally. Given the scale and scope of the problem, the global research on the subject is inadequate, and India-based research is almost non-existent. India and the world must address the challenge systemically and systematically. The ontology of geriatric disability care defines the system’s dimensions, elements, and pathways to deliver such care. The present global research on the subject studied through the lens of the ontology is sparse, skewed, and siloed. The paper presents an analysis of the emphases and gaps in the research and discusses its inadequacies to address the challenge. Last, the paper presents a roadmap for geriatric disability research in India to address the country’s challenge based on the ontology. The roadmap for research can be generalized to other countries and the globe. The global ontology can be used to integrate different local instantiations to accommodate and adapt to local conditions and requirements.

## Figures and Tables

**Figure 1 ijerph-19-10018-f001:**
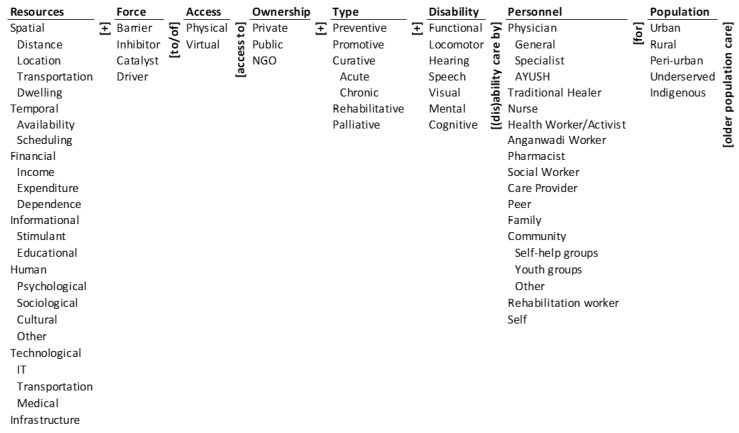
The ontology of barriers to and facilitators of access to geriatric disability care.

**Figure 2 ijerph-19-10018-f002:**
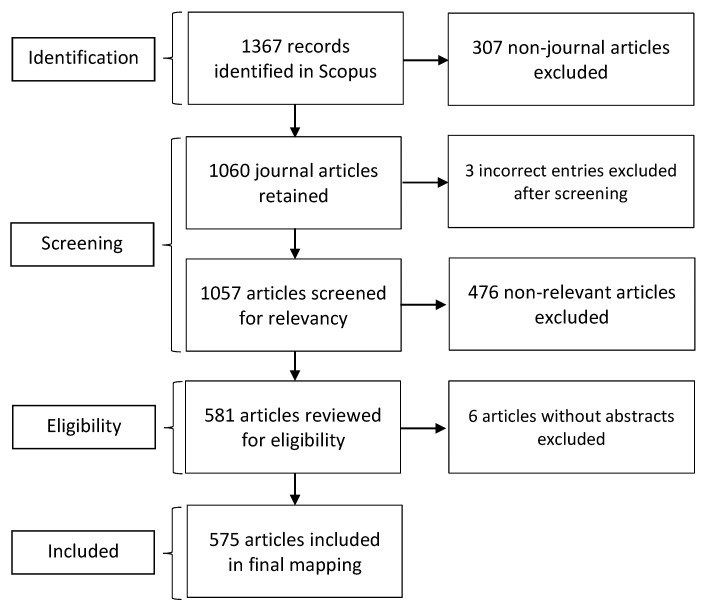
PRISMA diagram of corpus on access to geriatric disability care.

**Figure 3 ijerph-19-10018-f003:**
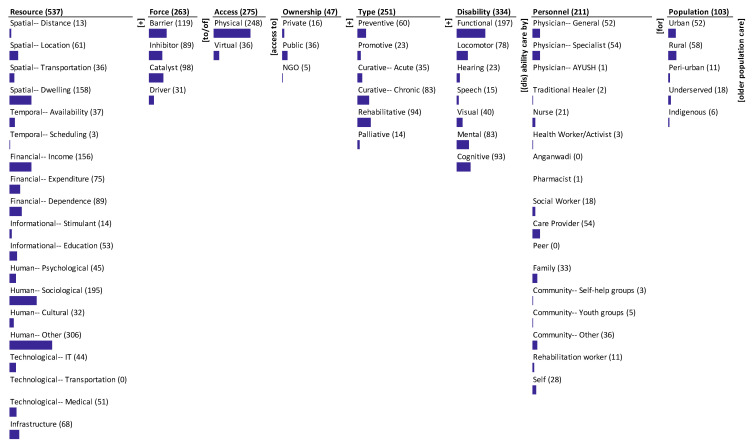
Monad map of access to geriatric disability care.

**Figure 4 ijerph-19-10018-f004:**
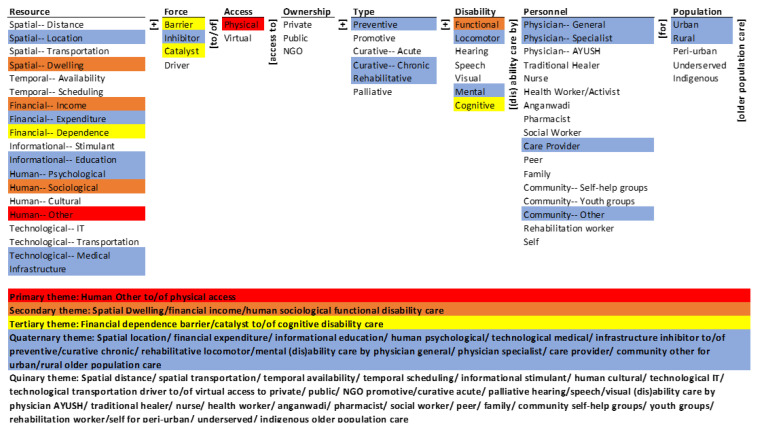
Theme map of access to geriatric disability care.

## Data Availability

Not applicable.

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
