# Peer review of "Access to Geriatric Disability Care in India: A Roadmap for Research"

_ijerph, 2022, doi:10.3390/ijerph191610018_

Round 1

Reviewer 1 Report

The topic is well thought of and required- particularly in the Indian context.

In the introduction, the authors may need to better elaborate on the need of this study. It may not be obvious for those not well versed with geriatric health research.

Similarly, in the methods section, the authors may want to start with a brief explanation of the Ontological methodology used and the terms used. 

A major limitation is the use of only ONE database for the source material- Scopus. Why were other medical lit database not included? A line on this may be added in discussion.

The main figure in the results seem to be the monash map. I would like the authors to explan the figure in a better way. It seems too complicated at first glance and takes effort to understand. But taht may be because I am not well versed with such maps-although most readers may be like me.

The discussion section is way too long. There is significant scope of reductiono and making it less verbose. The readers would lose interest if this section  is alomost as long as all the other parts combined. A better concise way may be found.

I think this article would be served well if they derive their own conceptual framework in terms of a simplified figure to illustrate their major findings and conclusions.

Author Response

Thank you for your kind comments. Please see the attached response.

Reviewer 2 Report

This is an interesting paper on geriatric disability care that suggests an ontology of access to geriatric care, utilizes the ontology as a framework for a review of global research on geriatric care and suggests on this basis a roadmap for geriatric research to address access to geriatric care in India. The paper is a valuable contribution to research on geriatric care in India and globally.

It is fair to use a global literature review to address the need for research in a context where research according to the author is "almost non-existent". Existing research is assumedly heavily "western" dominated, although this aspect (origin of existing research) is not sufficiently discussed in the paper. An Indian roadmap obviously needs to adapt to the context of India – and this point seems to be within the intentions of this paper. My suggestion is that this is made more explicit in a revision.

The India – global dimension in this paper creates a problem for the introduction which is very much concentrated on the Indian context. The authors need to bring in a global perspective in order to balance the text – for instance by starting the introduction with contextualising the topic in a global picture, i.e. the importance of the topic globally and references to international policies or global initiatives. The purpose or ambition of the paper, including the application of global research on developing the roadmap for India, should be mentioned as a logical conclusion of the first introductory paragraph. This can be followed by the current text related to India. Then return to the India – global dimension in the discussion and conclusion.

More specifically, there is reference to the situation being graver in rural than in urban areas in the introduction. Not entirely clear what this is based on. One can possibly assume that disintegration of family structures is primarily affecting urban areas and thus that the negative development (disintegration) would affect rural areas to a lesser degree. Of course, access to health care may be more problematic in rural areas but as the author makes a point out of the disintegration of family structures, it is suggested that this (urban – rural) is modified through an additional sentence or two – and preferably with reference to relevant literature.   

The ontological framework works well for the purpose of this paper. Although there is reference to some earlier published paper, it is nevertheless recommended to use a couple of sentences on how this was developed.      

Discussion: for some of the themes discussed there is consideration of relevance for India, for others not. The authors are asked to go through and add where this is missing.

There is need for a thorough read-through to edit the language. Small mistakes and awkward sentences here and there.

Author Response

Thank you for your kind comments. I hope you find the attached response satisfactory.

Reviewer 3 Report

I appreciate the effort of the authors. This is a well written manuscript. Please find my comments here:

1.      The authors have provided the Ontology of Barriers to and Facilitators of Access to Geriatric Disability Care in Figure 1. Please describe the process of the development of this. How the columns and domains under each column were selected? Did the author contextualize a priori framework?

2.      Authors only searched Scopus. This is a potential limitation. Please mention this in the limitations section.

3.      Please add a strength and limitations section.

4.      For the understanding of the reader, please define “Monad map” and “Theme map” at the beginning of the corresponding description.

5.      In methods, please describe the data synthesis process.

6.      The authors mentioned four themes in the theme map (primary, secondary, tertiary, and quaternary). Please describe how the themes were evolved/ generated? How the ranking (primary, secondary, and onward) was done? Authors have described as follows “Hierarchical cluster analysis was done using SPSS (Statistical Package for Social Sciences; IBM: Chicago, IL, USA) with simple matching coefficient (SMC) as the distance measure and the nearest-neighbor aggregation procedure. The detailed rationale for the choice of the clustering method and the presentation of the results are given in La Paz et al. [8] and Syn and Ramaprasad [9]. The five themes represent the five equidistant clusters in the dendrogram of the agglomeration [9].” However, it demands more clarity and explanation of the process.

Author Response

Thank you for your kind comments. We hope you find the attached response satisfactory.

Reviewer 4 Report

This paper is an ontological review on access to geriatric disability care. Although this is an important topic, there are several critical insufficiencies in this manuscript must be addressed.

#1 Overall

In this ontological review, the authors did not include “India” or its relevant key words for retrieving articles. It is not clear why the results of the study can “only” be adapted to India, as the title of the paper shows.

#2 Introduction, Page 2, Line 62

“This study maps the framework…...from our earlier publications on access to healthcare.”

>>> Please add references for the statement.

#3 Materials and Methods, 2.1 Ontology of barriers to and facilitators of access to geriatric disability care

There is no single source reference included in the section. It is not clear about the validity of the framework shown in Figure 1.

#4 Materials and Methods, 2.2 Method, Page 3, Line 119

It is not clear why the authors only use “Scopus”, instead of other databases, for retrieving articles. Multiple databases are commonly used in a review work, especially including Pubmed and some other major databases. By the way, running the same search on Pubmed, 3809 articles were hit (July 23, 2022).

#5 Materials and Methods, 2.2 Method, Page 3, Line 122

The final search was conducted more than one year before. Using the same database and the same terms for retrieval, 1558 articles (>>1367) were hit on July 23, 2022. Accordingly, the review must be updated.

#6 Materials and Methods, 2.2 Method, Page 3, Line 123

“(care AND access AND (geriatric OR elderly OR older) AND disability)”

>>>Terms included in the search are not comprehensive. For example, “access*” (with an asterisk) instead of “access” is commonly used to encompass “accessible”, “accessibility”, and so on. Also, synonyms of the key words are not included.

More, it is not clear how the framework (Figure 1) is connected to the search terms?

Author Response

Thank you very much for your critical comments. We hope you find the attached responses satisfactory. In addition to the ones in the attached document we have made a number of changes in response to the other reviewers that may have a bearing on your comments too.

Round 2

Reviewer 3 Report

I am happy with the changes made by the authors as per my previous comments.

Author Response

The reviewer has stated his/her happiness with the first set of revisions and has not sought further changes. Hence, there has been no additional changes in response to this reviewer.

Reviewer 4 Report

The final search was conducted on April 12, 2021, as the authors mentioned in the 2.2 Method section. Using the same database and the same terms for retrieval, 1558 articles (>>1367) were hit on July 23, 2022. Accordingly, the review must be updated.

Author Response

Reviewer 4

“The final search was conducted on April 12, 2021, as the authors mentioned in the 2.2 Method section. Using the same database and the same terms for retrieval, 1558 articles (>>1367) were hit on July 23, 2022. Accordingly, the review must be updated.”

We appreciate the reviewer’s concern with the currency of the corpus and effort to check it. The lag between the creation of the corpus and the publication of the results is inevitable, due to the logistics of the method and timelines of publication review. The time to download the corpus, curate it, code it, analyze it, and present the results is a major source of the lag. A second source of the lag is the time for submission, review, revision, rejection/acceptance.  In our publication experience of similar articles that have been cited in our paper (and others not cited), these lags are inherent in the method and the publication process.

In discussing the strengths and limitation in section 6, we have the following statement:

" A periodic review of the research using the framework could provide feedback to learn and redirect the trajectory of research, and consequently of policies and practices that translate the research.”

We have added the following to the above:

“As of July 23, 2022, 191 more articles had been published on the subject. Such additions can be cumulatively added to a subsequent analysis.”

A related but important question is the potential substantive impact of the new articles on the results and discussion. (Please note that many of the 191 new articles may be filtered by the inclusion/exclusion criteria.) In our unpublished analyses in two other domains, we have found minimal longitudinal changes, especially annually. Consequently, it would be reasonable to assume that the results with the additional articles would not be substantially different.

We hope the reviewer concurs with us.